

# Needs of family members of patients admitted to a university hospital critical care unit, Izmir Turkey: comparison of nurse and family perceptions

Sibel Büyükçoban[1], Zehra Mermi Bal[2], Ozlem Oner[1], Necmiye Kilicaslan[1], Necati Gökmen[1] and Meltem Ciçeklioğlu[3]

[1] Anesthesiology and Reanimation, Dokuz Eylül University, Balçova, Izmir, Turkey
[2] Intensive Care Unit, Düzce State Hospital, Merkez, Düzce, Turkey
[3] Public Health, Ege University, Bornova, Izmir, Turkey

## ABSTRACT

**Purpose:** . This study aims to compare the perceptions of nurses and families on the needs of the relatives of the patients in Intensive Care Unit (ICU).

**Methods**. This cross-sectional study was conducted in the ICU of a university hospital. The study comprised 213 critical care patients' relatives and 54 nurses working in the same ICU. Data were collected using the Turkish version of Critical Care Family Needs Inventory (CCFNI) and a questionnaire on the characteristics of the participants. The difference between the perceptions of families and nurses was analyzed using Student t-test. Results: CCFNI's assurance/proximity subscale mean scores ranked first among both patients and nurses. The item ''To be assured the best care possible is being given to the patient'' was the top priority for both groups. Mean assurance/proximity and information dimensions of relatives were significantly higher compared to nurses ($p < 0.001$). No significant difference was found between the perception of patient relatives and nurses related to support and comfort dimensions ($p < 0.05$).

**Conclusion**. The needs of the relatives of patients are underestimated by nurses. This inhibited the performance of ICU nurses in line with the holistic care approach. Educational objectives that include the needs of ICU patients' relatives should be incorporated into the undergraduate and in-service training of nurses. Policies should be established to create space and time for effective relative-nurse communication.

## INTRODUCTION

The stress level of the relatives of the patients who are admitted to Intensive Care Unit (ICU) is quite high due to serious and unstable conditions of their patients (*Khoshnodi, Masouleh & Seyed Fazelpour, 2017*; *Al Ghabeesh et al., 2014*). Moreover, as these patients are mostly unable to communicate due to sedation, mechanical ventilation, confusion, and coma, their family members are asked to make treatment decisions on the patient's

Corresponding author
Sibel Büyükçoban,
sibel.buyukcoban@deu.edu.tr,
sbuyukcoban101@gmail.com

behalf (*Khatri Chhetri & Thulung, 2018*). Procedures such as tracheotomy, operation consent, and transfer to the service can become very serious sources of conflict with the health care professionals at the point where the patient relatives have all decision-making rights. ICU nurses are in close contact with patients and their families, so they can support family members to overcome this process (*Al Ghabeesh et al., 2014*). In general, nursing practice plays a key role in the hospital setting. However, ICUs, where confusion and uncertainty prevail, are quite dynamic environments for nurse. This necessitates taking, fast and correct decisions. (*Mendonca & Warren, 1998*). Duties expected from a nurse under intensive care conditions may limit their ability to respond and support the needs of patient families when caring for intensive care patients (*Hetl et al., 2018*). The negligence contradicts with the holistic care approach, which is one of the professional features of the profession. Family participation, which is at the center of the holistic care approach, is an important component of the patient's treatment process (*Khatri Chhetri & Thulung, 2018*; *Al-Mutair et al., 2013*). Because the family effect is an important component in the patient's response to treatment, and nurses are medical personnel who best meet the emotional and social needs of families having patients treated in intensive care (*Hetl et al., 2018*). Organizational factors, work environment, nursing culture and the situation of the family affect the provision of family-centered services of nurses (*McAndrew et al., 2020*). Considering that patient-centered care is moving towards family-centered care in the provision of nursing services, it is very important to assess the needs of inpatient families, especially in intensive care units (*Khoshnodi, Masouleh & Seyed Fazelpour, 2017*).

Family members may experience extreme stress and anxiety, feel helpless and unable to cope with this situation (*Khatri Chhetri & Thulung, 2018*). Fear of death of their loved one, uncertainty about prognosis, financial concerns, changes in family roles, limited access to the critical care environment trigger feelings of shock, anger, denial, and despair within 72 h after admission to the ICU. They may even lead to feelings of guilt and depression in some cases (*Azoulay et al., 2005*; *Bijttebier, Vanoost & Delva D. Ferdinande P. Frans, 2001*).

Correct assessment of their needs is one of the first steps in providing appropriate health care to ICU patients and their families. *Leske (1986)* first described needs of the families of patients in critical care units under five dimensions; (1) support, (2) comfort, (3) information, (4) proximity, and (5) assurance (*Molter, 1979*; *Leske, 1991*). Nurses provide or coordinate requirements such as fulfilling the family's need for in these five dimensions through bedside family/patient interactions. Problems in understanding these family needs may make it difficult to cope with the crisis, which may eventually affect the patient's response to treatment (*Maxwell, Stuenkel & Saylor, 2007*).

Examining family members 'and nurses' perceptions of the needs of inpatient families in the ICU can provide an overview of the improvement of practices in this unit. Despite increasing evidence obtained from studies conducted in this area, the number of studies conducted in the Turkish society in the literature is very low (*Elay et al., 2020*; *Ozbayira, Tasdemir & Ozsekera, 2014*).

The socio-cultural and geographic contexts responsible for the diversity of family needs of ICU patients can be very important factors, so evidence from different cultures of the world is important (*Alsharari, 2019*). The objective of this study is to compare the

perceptions of nurses and families on the needs of the relatives of the patients in a university hospital Intensive Care Unit (ICU).

## METHODS

### Setting and samples

The present cross-sectional study was carried out at the Anesthesia Intensive Care Unit of Dokuz Eylül University Medical Faculty Hospital, one of the two major university hospitals in Izmir. The unit has a capacity of 13 beds, average staffing is two patients per nurse and annual patient capacity ranges between 450 and 500. The ICU, which provides tertiary-level intensive care, offers services to postoperative cases as well as patients who require mechanical ventilation for reasons such as polytrauma, chronic obstructive pulmonary disease, sepsis, and head trauma. Three days a week, medical information is given to the patient's relatives by a critical care physician. Afterwards, the attending nurse gives a bedside briefing on the day of the visit about the necessary materials and nursing care, and the questions of the patient's relatives are answered.

In this study, the average number of patients per year was accepted as a population size of 500, as only one family member was interviewed for each patient. The sample size representing the population was calculated as 278 using a 95% confidence interval, a 5% margin of error and unknown prevalence. A total of 213 family members of patients enrolled in the study, coverage rate was 76.6%. The study targeted all 57 nurses working in the same critical care unit, and 94.7% (n:54) coverage was achieved.

Inclusion criteria for patient families included.

(1) Age of 18 years or older

(2) Being a relative of the patient who signs informed consent form at hospital admission and related by kinship or marital relationship with the patient.

(3) visited the critically ill patient within 24 to 72 h after admission of the patient over 18 years of age in the ICU. Although the indications for hospitalization are very diverse, all of them are patients who are connected to a mechanical ventilator and whose relatives have been informed that they are in high danger of life.

(4) willing to participate in the study

(5) being able to read and write.

The inclusion criterion for nurses, on the other hand, was to be working in the Department of Anesthesiology and Reanimation Intensive Care Unit for at least six months. All nurses serve in the same working order alternately in 8-and 16-hour shifts.

### Data collection tools

The research data were collected using a questionnaire form, where the characteristics nurses and patients were questioned separately, and the Turkish version of Critical Care Family Need Inventory (CCFNI) (*Büyükçoban, Ciceklioglu & Demiral Yılmaz N. Civaner, 2015*). The questionnaires given to patient families included items questioning the age, gender, diagnosis of the patient as well as the age, gender, and relationship with patient. The questionnaires applied to nurses included the age, the duration of work in the ICU, and the experience of being a relative of a patient previously admitted to a critical care

unit. The Critical Care Family Need Inventory adapted to Turkish by Büyükçoban et al. was used in this study (*Büyükçoban, Ciceklioglu & Demiral Yılmaz N. Civaner, 2015*). The questionnaire developed by Leske comprised forty-five items that formed five major family "need" dimensions, namely, support (15 items), information (8 items), proximity or closeness (9 items), assurance (7 items) and comfort (6 items). Participants were asked to indicate the level of importance of each item measured on a 4-point Likert scale as follows; (1) Not important; (2) Slightly important; (3) Important; (4) Very important. Leske reported that the Cronbach alpha internal consistency coefficient calculated for the reliability study ranged between 0.61 and 0.88 for subscales and was 0.92 for the whole inventory (*Leske, 1991*). Unlike the original CCFNI, the revised Turkish version of the Inventory consists of fewer items (40) and three dimensions rather than five. Dimensions of the Turkish adaptation are described as 'support and comfort' (20 items), 'proximity and assurance' (11 items) and 'information' (nine items). The Cronbach alpha coefficient calculated for the internal consistency of the Turkish inventory is 0.93 for the whole inventory and between 0.83 and 0.92 for the sub scale (*Büyükçoban, Ciceklioglu & Demiral Yılmaz N. Civaner, 2015*).

In this study, Cronbach Alpha value of the scale was calculated as 0.89 for patient relatives and 0.95 for nurses.

The process of adapting the scale to Turkish was mainly carried out through relatives of patients, however, expert opinion from six intensive care nurses was received in assessment of the validity of the scope and then the expert panel included two intensive care nurses in addition to relatives of patients (*Büyükçoban, Ciceklioglu & Demiral Yılmaz N. Civaner, 2015*). In a study that used the Turkish version of CCFNI and included 50 nurses, the Cronbach Alpha value was found to be 0.90 (*Elay et al., 2020*). In addition, the Turkish version of CCFNI was applied to 8 intensive care nurses working in the cardiology intensive care unit of the same hospital prior to the study and the Cronbach Alpha value was found 0,96.

## Procedure

Ethical approval was granted by the Dokuz Eylül University Non-Clinical Studies Ethics Committee, and a research permit was obtained from the Head of the Intensive Care Unit of the Dokuz Eylül University Faculty of Medicine Department of Anesthesiology and Reanimation (IRB number: 2666-GOA. 2016/12-10, 05.05.2016). Data were collected between July 2018 and January 2019. Participants were verbally informed by an ICU physician about the objectives of the study and their written consent was obtained making clear that their participation would be on a voluntary basis. Confidentiality was preserved though anonymity of participants by refraining from questioning their names. Questionnaires were distributed to the participants by the same researcher, and they were asked to leave the completed questionnaires in the drop-off boxes placed in the waiting room. The research team explained the objectives of the study to the nurses, who were then given 30–35 min to fill in the questionnaires in-hospital during their break time. Eight patient relatives who failed to complete the questionnaires were excluded from the study.

### Data analysis

Data analysis was performed using SPSS 15.0 statistics package program. Independent $t$-test was used to evaluate the difference between perceptions of families and nurses. Whether the data indicated a homogeneous distribution checked using the Levene test, and in cases where it was not distributed homogeneously, the equal variations not assumed $p$ value was used. A $p$-value of $<0.05$ was regarded as statistically significant.

## RESULTS

The sociodemographic characteristics of the relatives and nurses are given in Table 1. Half of the members of patient families participated in the study were women, three quarters of them are in upper secondary education, six out of ten were children of ICU patients (Table 1). Overall, 6.6% of the relatives of the patients reported that they were in critical care units before and 45.1% of the relatives were previously admitted to ICU. When the patient characteristics were examined 60.6% of patients were male and 61.1% were over 65 years of age. Reasons for hospitalization among critical care patients were chronic obstructive pulmonary disease and post-operative care.

The mean age of the nurses was 31,9 ± 6,1 and %81,5 was female. Nurses with critical care experience of five years or more accounted for 35.2%. While 7.4% ($n = 4$) of the nurses reported that they were previously admitted to intensive care, 68.5% ($n = 37$) stated that at least one of their relatives was in an ICU.

A comparison of the mean item scores of patient families and nurses based on their answers to CFFNI items is shown in Table 2. Patient relatives gave the highest rank scores to items "To be assured the best care possible is being given to the patient," "To be called at home about changes in the patient's condition", and "To be assured it is alright to leave the hospital for a while". The needs ranked in the first and third places by the family members were equally important for the nurses. On the other hand, the mean score of patient relatives for both items were higher than that of nurses at a statistically significant level. The other two items perceived among the most important five needs for patient relatives were "To feel that the hospital personnel care about the patient" and "To know specific facts concerning the patient's progress". As for the nurses' perception of needs of patient relatives, the items ranked among the top five except for the two items cited above were "To have questions answered honestly" (second), "To know exactly what is being done to the patient" (fourth), and "To feel accepted by the hospital personnel" (fifth). Six of the ten most important needs ranked by patient relatives were also perceived among the top ten by nurses. Of the top ten needs perceived by relatives six items were related to assurance/proximity and four to information. Of the top ten needs of family members perceived by nurses seven items were related to assurance/proximity, two to comfort/support and one to information.

All the items in the last 10 among the needs of patient relatives were related to comfort and support subtitles. Also, there was no statistically significant difference between mean scores of family members and nurses in nine of these items.

There were statistically significant differences in 26 items in terms of scores provided by nurses and family members. Among these items, in 24 items that demonstrated significant

**Table 1  Sociodemographic characteristics of patients and relatives.**

| Characteristics | Number (%) |
|---|---|
| **Relative Characteristics** | |
| Gender | |
|     Female | 107 (50.2) |
|     Male | 106 (49.8) |
| Age | |
|     18–29 | 29 (13.6) |
|     30–39 | 42 (19.7) |
|     40–49 | 57 (26.8) |
|     50–59 | 55 (25.8) |
|     60 + | 30 (14.1) |
| Level of Education | |
|     Primary (5 year) | 15 (7.0) |
|     Secondary(8 year) | 39 (18.3) |
|     High School (12 year) | 67 (31.5) |
|     University + | 92 (43.2) |
| Relation with Patient | |
|     Child | 125 (58.7) |
|     Spouse | 28 (13.1) |
|     Parent | 18 (8.5) |
|     Sibling | 18 (8.5) |
|     Second-degree relative | 24 (11.3) |
| **Patient Characteristics** | |
| Gender | |
|     Female | 84 (39.4) |
|     Male | 129 (60.6) |
| Age | |
|     18–34 | 24 (11.3) |
|     35–49 | 18 (8.5) |
|     50–64 | 41 (19.2) |
|     65–79 | 86 (40.4) |
|     80 + | 44 (20.7) |
| Disease Diagnosis | |
|     Chronic obstructive pulmonary disease | 76 (35.7) |
|     Post-op care | 40 (18.8) |
|     Polytrauma | 30 (14.1) |
|     Cerebrovascular disease | 28 (13.1) |
|     Cardiovascular disease | 13 (6.1) |
|     Others | 26 (12.2) |

differences, the mean scores of family members were higher. The mean scores of nurses were found to be higher in "To be alone at any time" and "To be told about chaplain services" items where a significant difference existed between the scores of patient families and nurses (Table 2).

**Table 2  Comparision of the mean item scores of patient relatives and nurses.** Bold indicates *p* values smaller than 0.05.

| Dimensions | Items | Relatives (Rank) Mean± SD | Nurse (Rank) Mean ± SD | p |
|---|---|---|---|---|
| Asurance/Proxymity | To be assured the best care possible is being given to the patient | (1) 3.93 ± 0.26 | (1) 3.69 ± 0.46 | **<0.001** |
| Information | To be called at home about changes in the patient's condition | (2) 3.93 ± 0.26 | (12)3.35 ± 0.68 | **< 0.001** |
| Asurance/Proxymity | To be assured it is alright to leave the hospital for a while | (3) 3.93 ± 0.28 | (3) 3.52 ± 0.50 | **<0.001** |
| Asurance/Proxymity | To feel that the hospital personnel care about the patient | (4) 3.91 ± 0.29 | (6) 3.47 ± 0.58 | **<0.001** |
| Asurance/Proxymity | To know specific facts concerning the patient's progress | (5) 3.89 ± 0.32 | (8) 3.46 ± 0.50 | **<0.001** |
| Information | To know exactly what is being done for the patient | (6) 3.86 ± 0.36 | (4) 3.48 ± 0.57 | **<0.001** |
| Asurance/Proxymity | To have questions answered honestly | (7) 3.84 ± 0.39 | (2) 3.55 ± 0.54 | **<0.001** |
| Asurance/Proxymity | To feel there is hope | (8) 3.83 ± 0.37 | (13) 3.34 ± 0.58 | **<0.001** |
| Information | To know how the patient is being treated medically | (9) 3.83 ± 0.41 | (11)3.35 ± 0.56 | **<0.001** |
| Information | To know why things were done for patient | (10) 3.83 ± 0.41 | (14) 3.34 ± 0.65 | **<0.001** |
| Support/Comfort | To feel accepted by the hospital staff | (11) 3.80 ± 0.44 | (5) 3.48 ± 0.60 | **0.001** |
| Asurance/Proxymity | To receive information about the patient at least once a day | (12) 3.80 ± 0.45 | (25) 3.00 ± 0.75 | **<0.001** |
| Information | To be told about transfer plans while they are being made | (13) 3.80 ± 0.45 | (22) 3.17 ± 0.65 | **<0.001** |
| Information | To know about types of staff members taking care of the patient | (14) 3.76 ± 0.46 | (16) 3.31 ± 0.72 | **<0.001** |
| Asurance/Proxymity | To talk to the doctor every day | (15) 3.75 ± 0.46 | (20) 3.20 ± 0.71 | **<0.001** |
| Information | To have a specific person to call at the hospital when unable to visit | (16) 3.75 ± 0.54 | (27) 2.96 ± 0.87 | **<0.001** |
| Asurance/Proxymity | To know which staff members could give what type of information | (17) 3.71 ± 0.48 | (9) 3.46 ± 0.57 | **0.005** |
| Information | To talk about the possibility of the patient's death | (18) 3.55 ± 0.59 | (17) 3.30 ± 0.54 | **0.005** |
| Support/Comfort | To be told about other people that could help with problems | (19)3.55 ± 0.63 | (21) 3.17 ± 0.55 | **<0.001** |
| Asurance/Proxymity | To have directions as to what to do at the bedside | (20) 3.52 ± 0.59 | (15) 3.33 ± 0.78 | 0.111 |
| Asurance/Proxymity | To have explanations of the environment before going into the critical care unit for the first time | (21) 3.52 ± 0.69 | (7) 3.47 ± 0.69 | 0.671 |
| Support/Comfort | To see the patient frequently | (22) 3.50 ± 0.71 | (36) 2.70 ± 0.96 | **<0.001** |
| Information | To have visiting hours started on time | (23) 3.45 ± 0.59 | (18) 3.30 ± 0.60 | 0.087 |
| Support/Comfort | To talk about feelings about what has happened | (24) 3.41 ± 0.73 | (10) 3.36 ± 0.56 | 0.544 |
| Support/ Comfort | To have friends nearby for support | (25) 3.34 ± 0.68 | (19) 3.28 ± 0.59 | 0.553 |
| Support/Comfort | To talk to the same nurse every day | (26) 3.21 ± 0.73 | (39) 2.37 ± 0.94 | **<0.001** |
| Support/Comfort | To have another person with you visiting the critical care unit | (27) 3.21 ± 0.84 | (29) 2.91 ± 0.68 | **0.007** |
| Support/Comfort | To visit at any time | (28) 3.17 ± 0.82 | (40) 2.35 ± 1.01 | **<0.001** |
| Support/Comfort | To have a bathroom near the waiting room | (29) 3.13 ± 0.85 | (23) 3.04 ± 0.69 | 0.421 |
| Support/Comfort | To help with the patient's physical care | (30) 2.99 ± 0.84 | (38) 2.65 ± 0.91 | **0.016** |
| Support/Comfort | To have comfortable furniture in the waiting room | (31) 2.92 ± 0.97 | (30) 2.87 ± 0.95 | 0.738 |
| Support/Comfort | To feel it is all right to cry | (32) 2.88 ± 0.88 | (26) 2.98 ± 0.69 | 0.369 |
| Support/Comfort | To have good food available in the hospital | (33) 2.86 ± 0.96 | (24) 3.02 ± 0.86 | 0.279 |

**Table 2** (*continued*)

| Dimensions | Items | Relatives (Rank) Mean± SD | Nurse (Rank) Mean ± SD | p |
|---|---|---|---|---|
| Support/Comfort | To have a telephone near the waiting room | (34) 2.83 ± 1.01 | (33) 2.81 ± 0.99 | 0.920 |
| Support/Comfort | To be told about someone to help with family problems | (35)2.80 ± 0.93 | (31) 2.83 ± 0.75 | 0.758 |
| Support/Comfort | To have someone be concerned about with your health | (36) 2.78 ± 0.94 | (28) 2.94 ± 0.71 | 0.168 |
| Support/Comfort | To have a place to be alone while in the hospital | (37) 2.78 ± 0.96 | (35) 2.74 ± 0.83 | 0.793 |
| Support/Comfort | To be alone at any time | (38) 2.53 ± 0.85 | (34) 2.80 ± 0.81 | **0.038** |
| Support/Comfort | To have a pastor visit | (39) 2.37 ± 1.01 | (37) 2.65 ± 0.89 | 0.066 |
| Support/Comfort | To be told about chaplain services | (40) 2.35 ± 1.02 | (32) 2.81 ± 0.85 | **0.001** |

**Table 3   Comparision of mean subscale dimension scores of patient relatives and nurses.**

| | Relatives Mean (SD) | Nurse Mean (SD) | P |
|---|---|---|---|
| Support/Comfort | 3.03 (0.48) | 2.88 (0.55) | 0.060 |
| Assurance/Proximity | 3.79 (0.21) | 3.43 (0.36) | **<0.001** |
| Information | 3.76 (0.24) | 3.28 (0.41) | **<0.001** |
| Total Score | 3.40 (0.20) | 3.12(0.42) | **<0.001** |

The mean score on assurance/proximity subscale was ranked first by both family members and nurses. No statistical significance existed between family members' and nurses'perception of support/comfort needs. In terms of assurance/proximity and information, family members' mean perceptions of needs and mean total scale scores were found to be significantly higher than those of nurses (Table 3).

The experience of nurses when their own relatives have been in ICUs has been evaluated. A statistically significant difference in scale between nurses who had and had no experience of having a relative staying in intensive care was found for only one article. While nurses (2,51 ± 0,90) who had experience of having a relative staying in intensive care unit gave higher score for ''To talk to the same nurse every day'' article nurses (1,94 ± 0,85) having no such experience gave lower score (*p* : 0.035).

## DISCUSSION

This study evaluates the degree of coherence between the perceptions of the ICU nurses who assume the most important responsibility for fulfilling the needs of the patients as well as their families in ICU and the needs of patient relatives. It is very important to identify the difference between the perceptions of nurses and the needs of patients' relatives in the provision of family-centered nursing services in intensive care units.

Coherent with the literature, the results of this study showed that there are similarities and differences in terms of family members and nurses' perceived need for patient relatives (*Maxwell, Stuenkel & Saylor, 2007*; *Moggai, Biagi & Pompei, 2005*; *Hinkle & Fitzpatrick, 2011*; *Alnajjar & Elarousy, 2017*). Family scores were higher than those of nurses. This result is supported by the literature, which reports that nurses cannot adequately foresee the level of family needs and shows that the total quality of the services provided to

family members of critical care patients is an area that should be improved (*Gundo et al., 2014*).

Assurance and proximity subscales in the CCFNI developed by *Leske (1991)*, which reflect the need of the family to be physically and emotionally close to their critically ill family members, and have confidence in the patient's future, were rephrased under a single title as assurance/proximity in the Turkish version of the scale (*Büyükçoban, Ciceklioglu & Demiral Yılmaz N. Civaner, 2015*). Although the mean assurance/proximity subscale of family members was higher than that of nurses, it was ranked as the most important need in both groups. The compassionate and honest attitude of intensive care nurses can play an important role in meeting family members' e assurance and proximity need (*Maxwell, Stuenkel & Saylor, 2007*). Therefore, it is very important that the nurses in our work group be fully aware of the needs of family members in terms of assurance/proximity needs.

In many studies, assurance dimension was perceived as the highest priority need for both groups (*Gundo et al., 2014*; *Intessar, 2019*). In a literature review study, on the needs of family members, the assurance was found to be the top-ranked need regardless of the geographical region (*Padilla, 2014*). Furthermore, in the same study, the assurance subscale item "To ensure that the patient is being given the best possible care", which is determined as the most important need for North American families, was ranked eleventh in Asian families (*Padilla, 2014*). In our study, this was the item with the highest score both by nurses and family members. This item was ranked among the top five needs by both nurses and family members in comparative studies conducted in the US (*Hinkle & Fitzpatrick, 2011*), Belgium (*Bijttebier, Vanoost & Delva D. Ferdinande P. Frans, 2001*), Egypt (*Intessar, 2019*), Turkey (*Elay et al., 2020*) and Iran (*Shorofi et al., 2016*).

Although the mean score of family members was higher, the second most important need dimension reported by both groups was 'information'. In addition, the most basic needs of patient's relatives, such as knowing what was done to the patient, what kind of treatment was applied and why these treatments were performed and obtaining information about the changes in the patient's condition were not considered as priority by nurses. Moreover, the article titled "To be called at home about changes in the patient's condition" regarding information dimension was in the second and 12th rank for families and nurses, respectively. This finding showed that patients' need for information was not adequately perceived by nurses as described in the literature (*Bijttebier, Vanoost & Delva D. Ferdinande P. Frans, 2001*; *Al-Mutair et al., 2014*). In Turkey, especially information about treatment processes is mostly covered by doctors (*Ozbayira, Tasdemir & Ozsekera, 2014*; *Ünver, 2003*), which may have affected the perception of nurses.

Providing adequate information about the patient's condition, treatment and prognosis also fulfills the needs of families to trust the health system as well as healthcare employees (*Akhlak & Shdaifat, 2016*). To fulfill patient families', need for information, structured in-depth information tours rather than quick bedside conferences should be made, and nurses should be encouraged to get more actively involved in this process. Considering that this information to be provided to family members may be time consuming, there should be a sufficient number of personnel especially during visiting hours (*Bijttebier, Vanoost & Delva D. Ferdinande P. Frans, 2001*). In a study conducted in

Turkey it is also found that ICU nurses have problems communicating with the patient's family due to excessive workload, role conflict and uncertainty, environmental and institutional barriers (*Çınar, Olgun & Koyuncu, 2005*).

The "support and comfort" dimension of patient relatives' needs was perceived as the least important factor by both nurses and patients as described in the literature (*Padilla, 2014*; *Akhlak & Shdaifat, 2016*; *Bijttebier, Vanoost & Delva D. Ferdinande P. Frans, 2001*). In the initial days of emotional distress and continuous search for information, it seems logical that comfort factors have a low priority (*Bijttebier, Vanoost & Delva D. Ferdinande P. Frans, 2001*).

Considering the fact that the study was performed in single center where the level of education of patient relatives is relatively high, it should be noted that the generalizability of our results to the population in Turkey is limited. However, the present study similar to other studies in different ICUs in Turkey (*Elay et al., 2020*; *Ozbayira, Tasdemir & Ozsekera, 2014*) in terms of priority needs of the patients. The low number of nurses that participated in the study should be noted as another limitation.

## CONCLUSION

This study revealed that the most fundamental requirement for both patients and families was assurance and proximity in hospital staff. The second most important factor was the need for information that requires personal communication. However, in both dimensions, the fact that nurses' scores were lower than those of patient relatives indicates that nurses perceive the needs of patient relatives less than their actual needs.

Our study results support the evidence that Turkish version of CCFNI is a valid tool that allows evaluation of the family's needs and nurses' perception on these needs (*Büyükçoban, Ciceklioglu & Demiral Yılmaz N. Civaner, 2015*). However, as highlighted in the literature, considering the nature of the concepts involving these needs, the results need to be expanded and analyzed in depth using qualitative methods. For the nurses to fulfill such needs of the family members, it is quite important to define and assess these needs accurately. Family-centered care in the intensive care unit is defined as the assurance, emotional support, decision-making support provided by the nurse and acceptance of the family contributions to care (*McAndrew et al., 2020*).

In this framework, new strategies such as, flexible visiting hours, improved participation of nurses in 'visiting and "informing" hours and enhancing the quality of information / counseling processes for patient relatives should be addressed (*Bijttebier, Vanoost & Delva D. Ferdinande P. Frans, 2001*; *Padilla, 2014*). However, owing to staff shortages and excessive workloads, it makes it difficult for nurses to assume this role. For this reason, the health system as well as health institutions should create organizational conditions to support nurses (*Gundo et al., 2014*).

## ACKNOWLEDGEMENTS

We are grateful to the family members and nurses of the Intensive care unit for their participation and involvement in the study. We are grateful for the support of the following staff: Uğur Koca, Elvan Öçmen, Hale Aksu

### Funding

The authors received no funding for this work.

### Competing Interests

All authors declare there are no competing interests.

### Author Contributions

- Sibel Büyükçoban conceived and designed the experiments, performed the experiments, analyzed the data, prepared figures and/or tables, authored or reviewed drafts of the paper, and approved the final draft.
- Zehra Mermi Bal performed the experiments, analyzed the data, authored or reviewed drafts of the paper, and approved the final draft.
- Ozlem Oner analyzed the data, prepared figures and/or tables, authored or reviewed drafts of the paper, and approved the final draft.
- Necmiye Kilicaslan analyzed the data, authored or reviewed drafts of the paper, and approved the final draft.
- Necati Gökmen performed the experiments, prepared figures and/or tables, authored or reviewed drafts of the paper, and approved the final draft.
- Meltem Ciçeklioğlu conceived and designed the experiments, analyzed the data, prepared figures and/or tables, authored or reviewed drafts of the paper, and approved the final draft.

### Human Ethics

The following information was supplied relating to ethical approvals (i.e., approving body and any reference numbers):

Ethical approval was granted by the Dokuz Eylül University Non-Clinical Studies Ethics Committee, and a research permit was obtained from the Head of the Intensive Care Unit of the Dokuz Eylül University Faculty of Medicine Department of Anesthesiology and Reanimation (IRB number: 2666-GOA. 2016/12-10, 05.05.2016).

### Data Availability

Raw data are available in the Supplemental Files.

### Supplemental Information

Supplemental information for this article can be found online at http://dx.doi.org/10.7717/peerj.11125#supplemental-information.

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
