# Peer review of "Needs of family members of patients admitted to a university hospital critical care unit, Izmir Turkey: comparison of nurse and family perceptions"

_PeerJ, doi:10.7717/peerj.11125_

## Round 0.1 · original submission · Major Revisions

The manuscript is of interest and may deserve recommendation for publication once the Reviewers' comments are carefully addressed. Specifically, authors must provide an adequate rationale behind the use of a non-validated questionnaire in nurses.

·

Basic reporting

1. The authors include in their paper Table 3 that is not referenced throughout the text. This table provides summary information on the differences in means in each dimension of the questionnaire, and it may be useful to refer to it in the corresponding section of the text.

2. The bibliography used in the manuscript is adequate, but it is necessary to review the format in which it is reflected, both throughout the text and in the reference section.
Throughout the main text, the authors include a full stop between the end of the sentence and the reference number, to add another full stop after it. Authors should consider removing the first of these points. For example, in line 77, 98, 102, 105, and more.
In the reference section, there are several errors of style that can be amended to improve the manuscript. Considering that the authors are using the Vancouver referencing style, they should amend some errors such as avoiding underlining in the references (lines 308, 319, 341, and more), separating the names with commas and not points (lines 315 and 333), and avoiding using a colon after the authors' names (line 326). Also, in line 329 "MS" appears in the reference, and the authors should check if the surname is missing or those initials refer to other author. In line 343, the full name is used in one of the authors, and initials are used in the others. I would recommend that authors review the entire bibliography to ensure its correct formatting.

Experimental design

1. The authors use the Critical Care Family Needs Inventory to conduct their study. This questionnaire was validated in family members of people admitted to the ICU by Leske JS (Heart Lung 1991; 20(3):236-244) and its Turkish version was validated by Büyükçoban S et al (Peer J 2015;3:e1208. https://doi.org/10.7717/peerj.1208) also in relatives of persons admitted to ICU. However, the authors use this questionnaire in nurses, without pointing out any reference that this questionnaire is validated in that population. I would recommend the authors to include that reference if it exists, and if not, explain the relevance of using this questionnaire in that population.

2. The authors should consider explaining why they have taken into account whether the nurses had a history of ICU admissions. This aspect, a priori, may be an interesting factor because it may condition the nurse's perception of the needs of the family members, and a brief explanation of that importance in the paper would be appreciated.

Validity of the findings

Authors should consider including some Conclusions after the Discussion section, and these Conclusions should be according to the aims of the study.

Reviewer 2 ·

Basic reporting

The study of the biopsychosocial needs of ICU patients and their families is widely studied. Old references should not be used. I suggest updating the conceptual framework with current scientific literature.
It is strange that the introduction emphasizes the need for healthcare professionals to consider patients and relatives needs, but the study focuses on nurses. This needs to be justified as many of the aspects explored in the questionnaire are interdependent interventions.

Experimental design

The manuscript describes that the characteristics of the patients are quite heterogeneous. An elective postsurgical patient is not the same as a septic patient. The expectations and needs of the people screened can vary significantly depending on the circumstances of admission. The inclusion and exclusion criteria should address these issues, or at least explain how this has been controlled.
It is also necessary to evaluate not only the minimum time of admission to the ICU, but the accumulated time (the emotional exhaustion of a family that has been in the ICU for 20 days is not the same for a family admited for 2 days). Aspects such as the type of shifts that nurses work should also be valued. The type of family connection also needs to be better explained since the regular visitor is not the same as the sporadic.
Regarding the design, the time at which the study was carried out has not been indicated, which is very relevant although it is suspected that it was before the pandemic.
The size of both samples is very different and homogeneity tests would be necessary to verify that they are comparable.
Although the methodology used is widely explained in the methodology, it is necessary to make it clear throughout the text (including the abstract) that it is the Turkish version of the CCFNI.

Validity of the findings

I find the explanation of the results redundant since they are reflected in the tables. However, more discussion of such striking aspects is missed, such as the fact that the second important question for families is the twelfth for nurses. It is striking that honest information is valued more by nurses than by families.
It would be enriching if the relationship between the clinical variables of the patients and the needs of their families were analyzed. The current analysis without taking into account the context of the patients is of little use.
I believe that although the study responds to the initial objective, the manuscript is uninteresting without more complete accurate analyzes.

In the discussion again the problem referred to health workers is taken up, however the study has only been carried out on nurses. On the other hand, certain quite bold claims are made based on very local literature and with a limited level of evidence.

One of the most striking aspects are the recommendations made on the information. I do not believe it is fair to ask that nurses be involved in the process when it is a two-way process and the scientific literature has shown serious deficiencies in health teams, not only in nurses. Contextualization is necessary about why nurses do not consider giving information a priority. Is it their role in Turkey? Do they have the information requested? Is the information accessible?

Additional comments

In general, the study can provide the necessary knowledge to guide work and functions of ICU nurses in order to provide a better biopsychosocial care. The final paragraph of the manuscript is very important, in which the real situation of nurses is contextualized and I believe that this point should be present from the beginning (these circumstances may have an uncontrolled influence on the results of the study).

---

## Round 0.2 · accepted · Accept

The authors have adequately addressed the requirements and the final version has been improved over the original.

Reviewer 2 ·

Basic reporting

The authors have made substantial and sufficient improvements

Experimental design

The authors have made substantial improvements and have satisfactorily answered my questions. Thank you for clarifying the methodology used.

Validity of the findings

Although the study has limitations, I consider that the current version of the manuscript adequately collects the results and that they are valid.

Additional comments

Thank you very much for the improvements made and congratulations on the study. I would recommend that you continue with this line of research. It would be interesting to take a qualitative approach and involve more nurses in the research group.